# ATTENTION OVER PHRASES

## ABSTRACT

How to represent the sentence "That's the last straw for her"? The answer of the self-attention is a weighted sum of each individual words, i.e.

$$semantics = \alpha_1 Emb(\text{That}) + \alpha_2 Emb(\text{'s}) + \cdots + \alpha_n Emb(\text{her})$$

. But the weighted sum of "That's", "the", "last", "straw" can hardly represent the semantics of the phrase. We argue that the phrases play an important role in attention. If we combine some words into phrases, a more reasonable representation with compositions is

$$semantics = \alpha_1 Emb(\text{That's}) + Emb_2(\text{the last straw}) + \alpha_3 Emb(\text{for}) + \alpha_4 Emb(\text{her})$$

. While recent studies prefer to use the attention mechanism to represent the natural language, few noticed the word compositions. In this paper, we study the problem of representing such compositional attentions in phrases. In this paper, we proposed a new attention architecture called PhraseTransformer. Besides representing the words of the sentence, we introduce hypernodes to represent the candidate phrases in attention. PhraseTransformer has two phases. The first phase is used to attend over all word/phrase pairs, which is similar to the standard Transformer. The second phase is used to represent the inductive bias within each phrase. Specially, we incorporate the non-linear attention in the second phase. The non-linearity represents the the semantic mutations in phrases. The experimental performance has been greatly improved. In WMT16 English-German translation task, the BLEU increases from 20.90 (by Transformer) to 34.61 (by PhraseTransformer).

## 1 INTRODUCTION

The word-to-word attention (Bahdanau et al., 2015; Vaswani et al., 2017) updates the state of a word by the weighted sum of the current states of the words. The weights depend on the attention scores. Such mechanism incurs two major problems. Firstly, it is unfeasible to incorporate the semantics of phrases in the attention. For example, in figure 1a and figure 1b, using the attention over phrase "the last straw" is more intuitive than using the attention over the three individual words. However, such word composition (i.e. "the last straw") cannot be represented atomically by word level attentions. Secondly, the literal meaning of each individual words can be quite different from the implied meaning of the phrase. The weighted sum in attention cannot capture the semantic mutation. For example, it's hard to represent the semantics of the unpleasantness, the struggling, and being unable to continue by the weighted sum of individual words "the", "last", and "straw". Deepening the attention layers as in Devlin et al. (2019); Liu et al. (2019) cannot solve these issues.

Instead of representing the natural language via word-to-word attentions, incorporating phrase as atoms in the attention provides a more intuitive representation. In figure 1b, if we regard "the last straw" as an atom node in the attention in figure 1b, it is much more intuitive to represent the semantics.

For multiple words that form semantic compositions (e.g. idioms, chunkings, phrases), their semantics should be computed uniformly as an atom in the attention mechanism. The traditional word level attention mechanism should be expanded to represent the attentions over such compositions. To this end, we make a very simple framework called PhraseTransformer, based on Transformer. We leverage the concept of hypernodes in hypergraph. The traditional Transformer only consists of

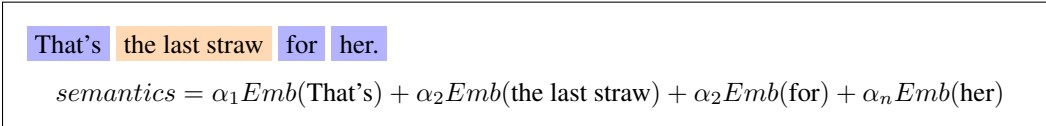

(a) Word level self-attention

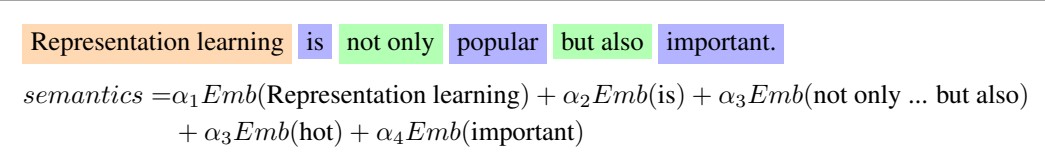

(b) Attention over adjacent phrases.

Representation learning   is   not only   popular   but also   important.

$$semantics = \alpha_1 Emb(\text{Representation learning}) + \alpha_2 Emb(\text{is}) + \alpha_3 Emb(\text{not only ... but also}) \\ + \alpha_3 Emb(\text{hot}) + \alpha_4 Emb(\text{important})$$

(c) Attention over nonadjacent phrases.

nodes representing single words. From the perspective of graph, Transformer is a complete bipartite graph, that the information transfers through all node pairs according to their attention scores. In hypergraph, a hypernode consists of one or more nodes. Likewise, we use such hypernodes in PhraseTransformer to represent the semantics of the composition. Each hypernode represents multiple words in a composition.

The information propagation in PhraseTransformer has two stages: the propagation across arbitrary node pairs and the propagation within a hypernode.

**Attention of arbitrary node pairs** works similar to the classical Transformer. We compute the attention scores for all nodes and all hypernodes. The information propagates accordingly. We use hypernodes to represent the atomic information propagation of semantic compositions in attentions.

**Non-linear attention within each phrase** reflects the inductive bias of the phrases. The nodes within a phrase should be updated by each other via attention. For example, the states of the hypernode `the last straw` should be updated by the node of `the`, `last`, `straw` etc. And the states of node `straw` should also be updated by the node of `the last straw`. To do this, we limit the attention in Transformer to nodes within each phrase.

Compared to the attention of arbitrary node pairs, a significant difference of the attention within each phrase is that we incorporate nonlinearity. As suggested in figure 1a, the literal meaning of each individual words can be quite different from the implied meaning of the phrase. We use the non-linearity to represent the semantic mutations of phrases.

Our proposed PhraseTransformer is able to capture any multi-word compositions. In this paper, we mainly study one of the simplest form of phrase: a sequence of adjacent words. For detecting phrases with adjacent words, we do not utilize existing identification algorithms (e.g. chunking). We roughly added all sequences of words in the sentence whose length is less than a given threshold. The attention mechanism itself decides the weights of these candidate phrases by assigning them different attention scores during the inference.

We summarize the contributions of this paper below:

- We propose to represent semantic compositions in the attention mechanism. We show that such representation is more intuitive than the traditional word-to-word attention.

- We incorporate the hypernodes in Transformer as PhraseTransformer. The semantics of a composition can be computed atomically via cross-node attentions. The inductive bias is reflected by the information propagation within each hypernode.

- We evaluate our approach over two widely used types of semantic compositions, i.e., chunking and semantic dependency. We conduct extensive experiments to verify the effectiveness of our proposed approaches.

## 2 MODEL

### 2.1 ARCHITECTURE

In this subsection, we show how the PhraseTransformer works on an abstract level. In the attention, we use "simple node" and "hypernode" to denote nodes that correspond to a single word and a phrase, respectively. We denote both the simple nodes and hypernodes as nodes. The PhraseTransformer contains $n$ simple nodes, where $n$ is the length of the sentence, and $m$ hypernodes representing $m$ phrases. For each node $s$, we denote the words it corresponds to as $word(s)$. For simple nodes, $|word(s)| = 1$. For hypernodes, $|word(s)| > 1$. For example, in figure 2, "straw" is represented by a single node $s_1$. $word(s_1) = $ (straw). The phrase "the last straw" is represented by a hypernode $s_2$, $word(s_2) = $ (the, last, straw).

The attention in PhraseTransformer has two phases. The first phase uses the node-to-node attention over all node pairs. This is represented by the blue edges in figure 2. For example, the state of the placeholder $[CLS]$ will be updated according to its attention scores to nodes of "That's", "for", "the last straw", etc. Similarly, the states of a hypernode (e.g. "the last straw") will be updated according to its attention scores to all nodes (e.g. "That's", "for", "last straw").

The second phase uses limited attention mechanism to update the states within each phrase. In figure 2, the orange edges show the attention within the phrase "the last straw". The state of $s_2$ will be updated according to its attention scores to "the", "last", "straw", but will not be updated by "for" or "her". And the state of $s_1$ will also be updated by $s_2$.

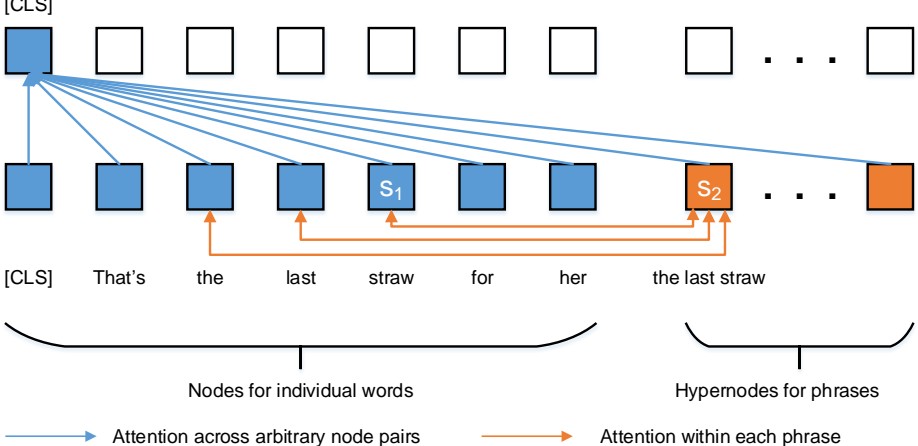

Figure 2: Architecture of PhraseTransformer.

### 2.2 PHRASE GENERATION

As mentioned in the introduction section, a hypernode represents a phrase formed by multiple words. In this paper, we consider the phrases formed by adjacent words in the original sentences. For a given length threshold $k$ and a word sequence $w_1, \cdots, w_n$, we generate hypernodes that consists of adjacent word sequence whose length is less than or equal to $k$. More specifically, we generate all hypernode $s$ such that

$$word(s) = (w_i, w_{i+1}, \cdots w_j), \text{s.t. } j - i < k \tag{1}$$

Our proposed neural network is feasible for any phrase generation strategies. We also compare with other the phrase generation w.r.t. the dependency parsing trees. The details of such strategy are shown in section 4.3.

## 2.3 MUTLI-HEAD ATTENTION OVER ALL NODE PAIRS

The attention over all node pairs in the first phase works similar to the standard Transformer (Vaswani et al., 2017). Given a sequence of vectors $H \in \mathbb{R}^{n+m}$, where $m$ is the number of hypernodes, we compute the attention of all nodes by:

$$Attn(Q, K, V) = softmax(\frac{QK^\mathsf{T}}{\sqrt{d}})V \tag{2}$$

, where $Q = HW^Q$, $K = HW^K$, $V = HW^V$, and $W^Q$, $W^K$, $W^V$ are trainable parameters. By following Transformer (Vaswani et al., 2017), we use multi-head attention to jointly attend to different representation subspaces:

$$MultiHead(H) = Concat(head_1, \cdots, head_h)W^O \tag{3}$$

, where $head_i = Attn(HW_i^Q, HW_i^K, HW_i^V)$, $h$ is the number of heads.

## 2.4 NON-LINEAR ATTENTION WITHIN EACH PHRASE

The attention within each phrase captures the inductive bias of the phrase. For example, we use the semantics of "representation", "learning" to update the semantics of phrase "representation learning", and vice versa. We consider two nodes have an edge between them if the words of one node is the subset of the words of another node. More specific, for two nodes $s_1$, $s_2$, we consider their attention within a phrase iff

$$word(s_1) \subseteq word(s_2) \text{ or } word(s_2) \subseteq Word(s_1) \tag{4}$$

. Given a sequence of vectors $H \in \mathbb{R}^{n+m}$, the attention within phrases are computed by:

$$Attn'(Q, K, V) = \sigma(softmax(\frac{QK^\mathsf{T}}{\sqrt{d}} \circ A)V) \tag{5}$$

where $Q = HW^{Q'}$, $K = HW^{K'}$, $V = HW^{V'}$, $\circ$ denotes the Hadamard product, $A \in [0, 1]^{(n+m) \times (n+m)}$ denotes the adjacency of the nodes:

$$A_{i,j} = \begin{cases} 1 & \text{if } word(s_1) \subseteq word(s_2) \text{ or } word(s_2) \subseteq word(s_1) \\ 0 & \text{otherwise.} \end{cases} \tag{6}$$

Note that we use an active function $\sigma$ in $Attn'(Q, K, V)$, which is not used in the standard Transformer. We found that the literal meaning of each individual words can be quite different from the implied meaning of the phrase. For example, in figure 1a, the semantics of "the last straw" is not a simple combination of the semantics of "the", "last", "straw". So using a weighted sum of the three words to represent the phrase cannot precisely represent its semantics. We use the non-linearity to represent the semantic mutation. In this paper, we use the sigmoid function as $\sigma$.

We also use multi-head attention in the attention within phrases to jointly attend to different representation subspaces::

$$MultiHead'(H, A) = Concat(head_1', \cdots, head_h')W^O \tag{7}$$

where $head_i' = Attn'(QW_i^{Q'}, KW_i^{K'}, VW_i^{V'})$.

## 2.5 ALGORITHM

The overall update algorithm of a $k$-layer PhraseTransformer is shown in Algorithm 1. We are given the initial states of the words $H \in \mathbb{R}^{n \times d}$. We first generate $m$ phrases and the adjacent matrix $A$. We extend the states of nodes to $H \in \mathbb{R}^{(n+m) \times d}$ by padding $m \times d$ zeros. Then we iteratively perform the attention over all node pairs and the attention within phrases $k$ times. The output is the states of the first $n$ nodes, which correspond to the original words of the sentence.

**Complexity analysis** The standard Transformer considers the attention over $n$ nodes. Thus its complexity is $O(n^2 d)$. Similarly, the first phase of PhraseTransformer (i.e. attention over all node pairs) is $O((n + m)^2 d)$, since it consider the attention over all $n + m$ nodes. The second phase of PhraseTransformer (i.e. attention within phrases) considers attentions within each phrase. Since the attention within phrases is sparse than attention over all nodes, overall complexity is $O((n + m)^2 d) = O(k^2 n^2 d)$. Such complexity is acceptable when $k$ is small. In the experiments, we set $k = 2$ or 3.

---

**Algorithm 1: The update algorithm of PhraseTransformer**

  **Data:** $H, n$;
  //Generate the hypernodes and init the states to zeros;
**1** Generate $m$ hypernodes and $A$;
**2** $H_0 \leftarrow$ Pad $m \times d$ zeros to $H$;
**3** **for** $i = 1...k$ **do**
    //Attention over all node pairs;
**4**    $H_i \leftarrow MultiHead(H_{i-1})$;
    //Attention within phrases;
**5**    $H_i \leftarrow MultiHead'(H_i, A)$;
**6** **return** $H_k[: n, :]$

---

## 3 RELATED WORK

**Attention-based Natural Language Representation** There is a trend to use attention and self-attention Bahdanau et al. (2015); Vaswani et al. (2017) for modeling the natural language. Attention is able to jointly model the semantics of the words and their alignments Bahdanau et al. (2015) or compositions Vaswani et al. (2017). It captures compositions of all word pairs, which is superior to traditional sequential representation (e.g. LSTM) in many tasks Wang et al. (2018). However, previous attention mechanisms work over the word level. They are infeasible to capture the semantics of phrases, especially if the semantics of the phrase is different from the literal semantics of its words. Instead, our proposed PhraseTransformer solves such issue by introducing hypernodes to represent phrases in the attention.

Attention-based pre-training language models achieve state-of-the-art results in many NLP tasks. BERT Devlin et al. (2019), GPT-2 Radford et al. (2019), Roberta Liu et al. (2019) use multi-layer Transformers as the major network structure. We believe that our proposed PhraseTransformer is a good alternative of Transformer for pre-training models, because using phrase-based knowledge is more intuitive than the word-based knowledge. On the hand, as PhraseTransformer has more trainable parameters than Transformer, the large amount of training data in pre-training guarantees these parameters can be trained sufficiently.

**Incorporating Inductive Bias in Natural Language Representation** is another trend in NLP. These inductive bias are usually from semantic dependencies Strubell et al. (2018); Zhang et al. (2018); Fu et al. (2019) or other external knowledge. LISA Strubell et al. (2018) introduce the dependency parsing as inductive bias to Transformer. It only allows the information transfer through edges of the parsing tree. C-GCN Zhang et al. (2018) and GraphRel Fu et al. (2019) uses graph convolution networks, in which the edges are from the dependency tree, to represent such inductive bias. These approaches all relies on a good algorithm (e.g. dependency parsing) for the given domain. This is usually unsatisfied for different languages and different domains. In contrast, our approach do not rely such knowledge to represent the compositions.

## 4 EXPERIMENTS

In this section, we evaluate our proposed approaches over two classical NLP tasks: machine translation and POS tagging. All the experiments run over a computer with Intel Core i7 4.0GHz CPU, 96GB RAM, and a GeForce GTX 2080 Ti GPU.

### 4.1 MACHINE TRANSLATION

**Hyper-parameters** We set the dimension of hidden states as 512, the number of heads as 8, and the dimension of each head as 64. We set the number of layers to 6.

**Dataset** We use the WMT'16 multimodal translation: Multi30k (de-en), which consists of 29k samples for training, 10k samples for development, and 1k samples for testing.

**Neural network details:** Each layer of the Transformer has two self-attention modules: one for the encoder, and one for the decoder. We apply the PhraseTransformer to the self attention module in the encoder layer.

| Model | BLEU |
|---|---|
| Transformer | 20.90 |
| Linear PhraseTransformer (k=2) | 21.60 |
| Linear PhraseTransformer (k=3) | 20.95 |
| PhraseTransformer (k=2) | **34.61** |
| PhraseTransformer (k=3) | 34.03 |

Table 1: Results over WMT'16 multimodal translation.

The results of our proposed approaches are shown in table 1. The PhraseTransformer outperforms Transformer by a large margin. This shows that the intuition of representing natural language via phrases works.

**Effectiveness of the non-linearity** We want to figure out what make the PhraseTransformer works so well. Note that PhraseTransformer uses an non-linear function in the attention within phrases. We further verify the effectiveness of such non-linearity by using a linear competitor, which is denoted as `linear PhraseTransformer`. It works the same as PhraseTransformer, except that it does not use the activate function in equation equation 5. We report the results of the linear PhraseTransformer in table 2.

The comparison clearly suggests that most improvement of the PhraseTransformer is obtained from the non-linear activation. This is intuitive because we want to use the non-linearity to represent the semantic mutation in phrases.

### 4.2 Visualization of PhraseTransformer

We visualize the attention matrix of PhraseTransformer in the self-attention module for WMT'16 multimodal translation in figure 3. We analyze the attention over these phrases. As can be seen, the attention is over the single nodes and the hypernodes. This verifies that the PhraseTransformer successfully use phrases as atoms in attention. For example, in figure 3a, we can see that the phrase "a building" has a higher attention score than other phrases. This makes sense because the two words form a valid phrase. In the attention matrix for attention within phrases in figure 3b, we found that the word "on" attends to the phrase "working on". This is intuitive because the work "on" itself does not have a clear semantics. It needs the collocation with "working" to represent its semantics.

### 4.3 POS tagging

**Hyper-parameters** We follow the parameters in Guo et al. (2019). We set the dimension of hidden states as 300, the number of heads as 6, and the dimension of each head as 50. We set the number of layers to 2.

**Dataset** We use the POS tagging dataset from Penn Treebank (PTB) Marcus et al. (1993), which consists of 38k samples for training, 5k samples for development, and 5k samples for testing.

**Competitor with semantic inductive bias** Noticing that the semantic dependency is also a possible inductive bias (Strubell et al., 2018; Zhang et al., 2018), we also compare with PhraseTransformer with dependency parsing trees. We denote it as semantic PhraseTransformer. Each hypernode in it represents an edge in the dependency parsing tree. In this way, the non-adjacent words can form one phrase. For example, it is possible to represent the phrase "not only ... but also" in figure 1c. More formally, in semantic PhraseTransformer, we generate hypernode $s$ such that

$$word(s) = (w_i, w_j) \text{ s.t. edge } (w_i, w_j) \text{ is in the dependency parsing tree} \tag{8}$$

. In this paper, we use spaCy (Honnibal & Montani, 2017) for dependency parsing. The two phases in the semantic PhraseTransformer works the same as in PhraseTransformer.

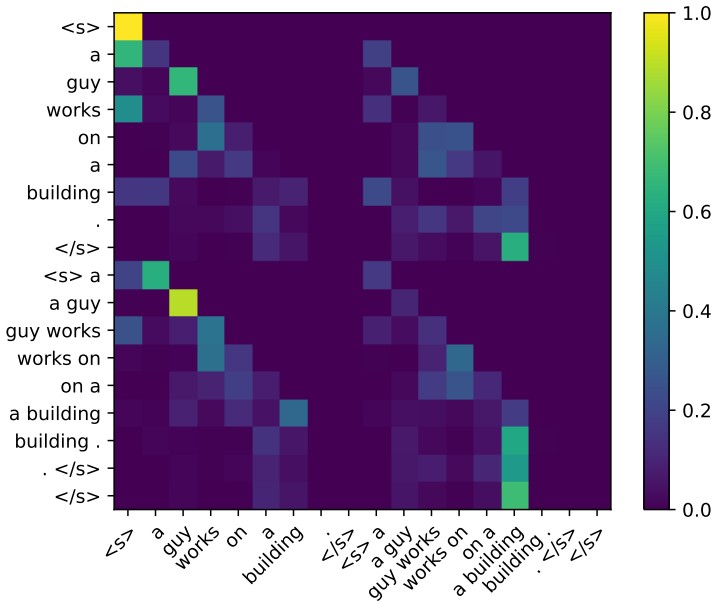

(a) Attention matrix for attention over all nodes.

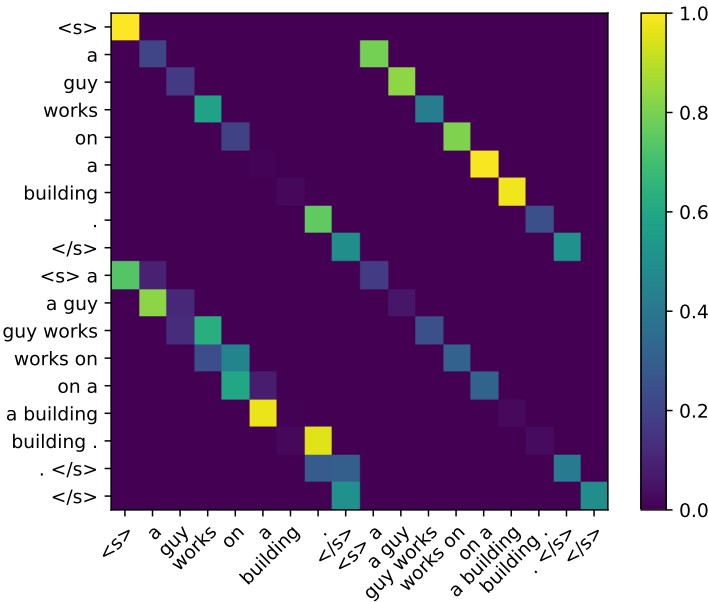

(b) Attention matrix for attention within phrases.

Figure 3: Attention matrix visualization.

The results are shown in table 2. The PhraseTransformer outperforms Transformer and semantic PhraseTransformer. Note that semantic PhraseTransformer uses external data to incorporate inductive bias. This verifies we incorporate simple but suitable inductive bias, i.e., the phrases.

## 5 CONCLUSION

We noticed that representing phrase as the atoms in attention is more intuitive than the word-to-word attention. Wee proposed the PhraseTransformer, which incorporates the hypernodes in Transformer. Each hypernode represents a possible phrase in the sentence. Without using extra phrase identifi-

| Model | use external data | accuracy |
|---|---|---|
| Transformer | | 96.36 |
| Semantic PhraseTransformer (k=2) | ✓ | 96.35 |
| Semantic PhraseTransformer (k=3) | ✓ | 96.23 |
| Linear PhraseTransformer (k=2) | | 96.54 |
| Linear PhraseTransformer (k=3) | | 96.52 |
| PhraseTransformer (k=2) | | **96.87** |
| PhraseTransformer (k=3) | | 96.80 |

Table 2: Results over PTB POS tagging.

cation algorithms, we jointly detect the phrases and attend over them. Our proposed PhraseTransformer has two phases. The first phase achieves phrase level attentions like the word level attention. The second phrase represents the inductive bias between words and phrases. The experiments show a significant improvement in the machine translation task.

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
