# OpenReview forum: "Attention over Phrases"
_ICLR.cc/2020/Conference — Reject_

### Official Review · AnonReviewer1 · 2019-10-24
**Official Blind Review #1**

**Rating:** 3

**Review:**

This paper addresses an issue of compositionality in self-attention models such Transformer. A simple idea of composing multiple words into a phrase as a hypernode and representing it using a non-linear function to capture the semantic mutation is proposed. In the machine translation and PoS tagging tasks, the proposed PhraseTransformer achieves impressive gain, especially +13.7 BLEU score compared to the Transformer.

The motivation of the paper is very clear, and I love this kind of paper; with a simple idea, making a huge impact on the field. I appreciate the real example to compare how word-level self-attention is different from the phrase-level self-attention in Abstract and Figure 1. The problem itself; tackling the semantic compositionality of self-attention, is a very important problem, and I like the part that authors described it as an inductive bias as a model perspective.

However, this work seems to be problematic in terms of presentation, clarity, and meaningful comparisons. Please see my detailed comments below.

First, what exactly is “semantic mutation”? The term has been used here and there to describe the inductive bias in semantic compositionality and to show how the nonlinearity can effectively capture it. But, I couldn’t find any definition from the paper, couldn’t find any formal definition from any ACL papers, and couldn’t guess myself based on the context. I am guessing it is probably some sort of combination of meaning in phrase-level words. If so, more importantly, how does the simple non-linear function (i.e., sigmoid) can capture such semantic combinations of words? How could it make such a huge gain (PhraseTransformer vs Linear PhraseTransformer) in Table 1 on MT task? This seems to be the most important contribution of this paper, of which I don’t understand yet.


What is the “concept of hypernodes in hypergraph”? I think it is not a common word that I can understand it clearly without any references or any background. It would be better to add some references for the concept. Again, I am guessing it is a sort of graph theory that decomposes a large node into small pieces but keeps their connectivity. But, then how is that exactly linked to phrases of words? If you make phrases only on consecutive words, it is basically just chunking. I don’t find any relevance of phrases in a sentence with the (hyper)graph something.

In Figure 2, how is the bidirectional path made between the word representations and phrase representations? In my understanding of your algorithm based on Equation 2-6, I only see the attention of phrases is computed by word attention, but not the other way.  Please clarify this. If so, how does the gradient back-propagate to each other?

The biggest concern of this work is the scores reported in Table 1. I have checked the recent papers which used the Mutli30K(de-en) and other results from WMT[16-18] reports, but the BLEU score reported in Table 1 (20.90) seems way lower than the scores reported by any systems trained by either non-Transformer or Transformer systems. For that reason, it would be fair to include some results from the state-of-the-art systems on the same dataset.

A minor point but in the complexity analysis, your m is basically n because you take consecutive words from n length of sentence. You better distinguish which variables are dependent on each other first.


There are MANY typos, missing captions, grammatical errors in the paper. Here are only some of them:
There is no caption for Figure 1.
When you cite a reference with its model name, you better make the citation under parenthesis.
“equation equation 5” -? “equation 5”
“Wee” -> “We”


**Experience Assessment:**

I have read many papers in this area.

**Review Assessment: Checking Correctness Of Derivations And Theory:**

I assessed the sensibility of the derivations and theory.

**Review Assessment: Checking Correctness Of Experiments:**

I assessed the sensibility of the experiments.

**Review Assessment: Thoroughness In Paper Reading:**

I made a quick assessment of this paper.

---

### Official Review · AnonReviewer2 · 2019-10-24
**Official Blind Review #2**

**Rating:** 1

**Review:**

This submission proposes to consider to put attention on "phrases" in NLP. The phrases are generated by taking consecutive words in sentences. Each phrase is treated as a "node" in the same way as words. Then representations of phrases are learned in the network. The algorithm is applied to two applications, translation and pos tagging. The proposed method achieved better performance than transformer.

Critics:

1. In the abstract and the introduction, the submission argues that usefulness of phrases, which are sementic units represented by word groups. However, in the model development, "phrases" are really bigrams and trigrams. I don't know how much the previous argument is still valid. Particularly, there are so many bigrams and trigrams. The effect from these word combinations should have strong effect on the model, but the effect may not be explained as the argument.

2. I think transformer can somewhat capture word combinations in bigrams and trigrams. In higher layers, transformer actually combine words in representations. What is the advantage of the proposed method over the type of combination done in transformer?

3. The experiment only compares to transformer in the translation task. It only compares to transformer and semantic phrase transformer. Other SOTA methods (e.g. different versions of transformers) should be compared.

4. The comparison is not really fair. In each "layer" of the proposed phrase transformer, it has actually two self-attention layers, but the baseline has only one self-attention layer.



**Experience Assessment:**

I have published in this field for several years.

**Review Assessment: Checking Correctness Of Derivations And Theory:**

I carefully checked the derivations and theory.

**Review Assessment: Checking Correctness Of Experiments:**

I assessed the sensibility of the experiments.

**Review Assessment: Thoroughness In Paper Reading:**

I read the paper thoroughly.

---

### Official Review · AnonReviewer3 · 2019-10-29
**Official Blind Review #3**

**Rating:** 1

**Review:**

I think the paper needs a deep review in the English part. For example, in the abstract, they repeat "In this paper" a couple of time and it is complicated to understand the introduction and methodology. Also, I think the paper needs a better structure. The related work should be first in order to understand the relevance of this paper.

From the experiment point of view, it is necessary a better explanation about the hyperparameters or the experiments which were carried out. In addition, the single database was used to evaluate the technology which is not enough to show the big different respect to the transformed paper. Also, the comparison is not really fair. In each "layer" of the proposed phrase transformer, it has actually two self-attention layers, but the baseline has only one self-attention layer. In addition more methodologies should be necessary to compare the results of the experiment.

The architecture part is complicated to follow and I don't understand the big contribution of this paper.

For that reason, I recommend a reject the paper and work more for the final version

**Experience Assessment:**

I have published one or two papers in this area.

**Review Assessment: Checking Correctness Of Derivations And Theory:**

I assessed the sensibility of the derivations and theory.

**Review Assessment: Checking Correctness Of Experiments:**

I assessed the sensibility of the experiments.

**Review Assessment: Thoroughness In Paper Reading:**

I read the paper at least twice and used my best judgement in assessing the paper.

---

### Public Comment · ~Sai_Prabhakar_Pandi_Selvaraj1 · 2019-10-01
**Interesting approach**

I liked the overall idea and implementation.

There are couple of typing mistakes in the paper:
1. In the overall complexity formula on page 4, I think '=' should be replaced with '+'
2. In section 2.3 first paragraph, H \n R^{(n+m)xd} instead of H \n R^{(n+m)}
3. wee -> we

It would be awesome if you can also show how the hypernode attention helps via visualizations or reduction in errors dealing with phrases.
Can you comment on the training time changes?

---

### Public Comment · ~Hao_Zhang1 · 2019-10-03
**Results on your table 1**

A good idea to add phrase information in the transformer.
However, I have questions about your experiments.

I think your results on table 1 for transformer is not good. I run the transformer using OpenNMT code on Multi30K dataset, achieving the Blue value: 34.2, without adjust hyper parameters seriously. So I want to know why you report 20?

The second question is that why do not perform your experiments on standard dataset such as wmt14' or more languages as done on "attention is all your need"?

Thanks.

---

> ### Author Response · Authors · 2019-10-05
> **Detailed results by OpenNMT**
>
> Thanks for your comments.
>
> According to your first comment, we tried to run the Transformer using OpenNMT over the Multi30k dataset, and got BLEU score of 23.47. Could you specific the detailed hyper-parameters you use in OpenNMT? We are glad to see a better baseline model that you implemented, which may further verify the effectiveness of our method if we apply our optimizations over such a baseline.
>
> By following the parameters in <Attention Is All You Need>, currently we use the command below:
> onmt_train -encoder_type transformer -decoder_type transformer -gpu_ranks 0 -enc_layers 6 -dec_layers 6 -heads 8 -rnn_size 512 -src_word_vec_size 512 -tgt_word_vec_size 512 -learning_rate 0.001 -optim adam -adam_beta2 0.98
>
> As to the second comment, we are trying to add more experimental analysis. It may take several days.

---

### Public Comment · ~Hamed_GHAZAVI_KHORASGANI1 · 2019-10-21
**k-2 vs k-3 Perfromance**

Very interesting work! I think the authors should explain why in all the experiments (Table 1 and Table 2) PhraseTransformer (k=2) outperforms PhraseTransformer (k=3)? If attention over phrases is the answer, shouldn't we get better results with higher k?

---

> ### Author Response · Authors · 2019-10-25
> **Response**
>
> Thanks for your comments. Very interesting question.
>
> One possible answer is, higher k means more parameters, which leads to overfitting.
>
> We further investigate such phenomenon in PhraseTransformer. Another possible explanation is, as we are using a #deep# neural network, the information of the 2 word composisions in lower layers will propagate to more complicated compositions in higher layers. So we do not need to explicitly use higher k to represent longer phrases. To verify this, we investigate the effectiveness of our proposed method for n_layer=1. In this case, there is no such information propagation. We obtained the results below. In this setting, PhraseTransformer(k=3) performs slightly better than PhraseTransformer(k=2). This is consistent with our above explanation.
>
>  model                                                             BLEU
>  ══════════════════════════════════════
>  PhraseTransformer(k=2,n_layer=1)            29.94
>  ──────────────────────────────────────
>  PhraseTransformer(k=3,n_layer=1)            30.14

---

> > ### Public Comment · ~Hamed_GHAZAVI_KHORASGANI1 · 2019-10-29
> > **How about attention over words, does that propagate to more complicated compositions in higher layers as well?**
> >
> > Thank you so much for your comprehensive response. If we accept the second explanation ("the information of the 2 word compositions in lower layers will propagate to more complicated compositions in higher layers"), can we also say the information related to phrases (whether 2, 3 or more) would be implicitly captured in higher layers when we use normal attention over words. If this is the case, attention over phrases is only necessary when we don't have enough data or computational resources for extensive training.

---

> > > ### Author Response · Authors · 2019-10-31
> > > **Response**
> > >
> > > I think they are different. The information related to phrases in our approach cannot be captured in higher layers in normal attention over words. For k=2, our approach represents the inductive bias and non-linearity of phrases with length 2. For "deep" network structure, such inductive bias and non-linearity of 2-word phrases propagate to more complicated phrases. But in normal attention over words, neither the inductive bias nor the non-linearity are captured. So we cannot expect the similar performance of the normal attention over words if we have more data or deeper structures.

---

> > > > ### Public Comment · ~Hamed_GHAZAVI_KHORASGANI1 · 2019-10-31
> > > > **Thank you for your response**
> > > >
> > > > Your answer makes perfect sense as the non-linearity is proposed by your method and does not exist in normal attention.  Thanks again for the clarifications.

---

### Decision · Program_Chairs · 2019-12-19

**Decision:**

Reject

**Comment:**

This paper incorporates phrases within the transformer architecture.

The underlying idea is interesting, but the reviewers have raised serious concerns with both clarity and the trustworthiness of the experimental evaluation, and thus I cannot recommend acceptance at this time.